# Implicit Reward Alignment for Causally Coherent Tabular Data Generators

## Abstract

Foundation models for structured data are increasingly being used as queryable generators for scenario planning and counterfactual analysis, requiring models that are statistically realistic, causally coherent, and capable of conditional querying from partial inputs. Yet existing approaches to tabular data generation either optimize solely for distributional fidelity, or impose causality through explicit structural assumptions, which are untenable in a real-world setting. We argue that **counterfactual queryability** is a key missing reliability axis, for tabular FMs and introduce Causal Reward Aligned Fine-Tuning (CRAFT), a reinforcement learning framework where language models are trained to produce causally-consistent samples only through implicit reward signals. Across multiple semi-synthetic and benchmark settings, we compare against 15 baseline models spanning LLM-, diffusion-, VAE-, and GAN-based generators, using multiple metrics along both realism and causal coherence dimensions, and show that reward-based alignment improves counterfactual accuracy and treatment-effect estimation, even in settings where baseline models achieve comparable distributional realism. More broadly, these findings suggest that intervention-consistent generative behavior can emerge from alignment objectives alone, without strong architectural priors or explicit encoding of the causal graph.

## 1. Introduction

Many high-stakes applications, such as treatment selection, pricing, and policy design, depend on the ability to perform counterfactual reasoning about potential outcomes. Synthetic tabular data offers a promising mechanism for simulating scenarios by generating samples that resemble observed populations. However, our experiments show that distributional realism is insufficient to guarantee causal consistency, which is crucial for counterfactual reasoning. To support counterfactual reasoning, generative models must satisfy two additional requirements: **(i) conditional queryability**, the ability to generate samples conditioned on partial inputs, and **(ii) causal consistency**, the ability to produce intervention-consistent outcomes.

Existing tabular generators occupy three incomplete regimes (see Table 1). Realism-driven models such as GAN-, VAE-, and diffusion-based synthesizers match observational distributions but do not natively support counterfactual querying. LLM-based tabular generators are queryable through partial-row completion, but are trained for likelihood or utility rather than interventional consistency. Causally explicit generators encode structural assumptions, but require specifying or learning a causal graph (see Appendix A.1).

| Category | Conditional Queryability | Causal Optimization | Model-free Generation |
|---|---|---|---|
| *Realism-Driven* | $\sim$ | $\times$ | $\checkmark$ |
| *Queryable* | $\checkmark$ | $\times$ | $\checkmark$ |
| *Causally-Explicit* | $\checkmark$ | $\checkmark$ | $\times$ |
| **CRAFT** | $\checkmark$ | $\checkmark$ | $\checkmark$ |

*Table 1.* Comparison of tabular generation paradigms.

The missing combination creates a practical gap: decision-makers need generators that are simultaneously realistic, queryable, and causally coherent, without requiring full specification of the underlying data-generating process. We seek to establish the following:

1. Can a generative model be made causally consistent purely through reward signals, without explicit structural modeling, while preserving realism?

2. Can causal rewards estimated from data suffice when the true data-generating process is not accessible?

We emphasize that our goal is not structure recovery or identification of structural parameters, which is generally infeasible without additional assumptions such as multiple

[1]Anonymous Institution, Anonymous City, Anonymous Region, Anonymous Country. Correspondence to: Anonymous Author <anon.email@domain.com>.

Preliminary work. Under review by the International Conference on Machine Learning (ICML). Do not distribute.

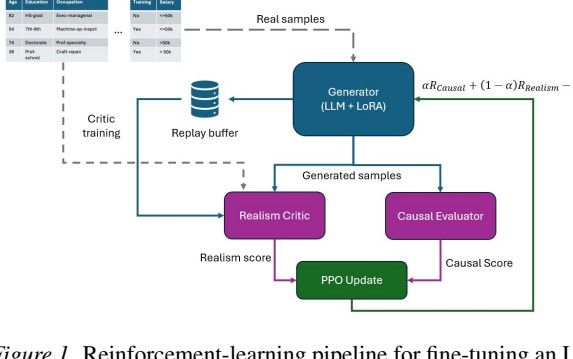

*Figure 1.* Reinforcement-learning pipeline for fine-tuning an LLM-based tabular generator using both realism and causal-coherence rewards.

environments or interventional data (Varici et al., 2024). We instead target interventional consistency at the sample level: the causal reward shapes the generator to respond appropriately to interventions, even when the underlying causal model is only partially identifiable.

To answer these questions, we train a queryable generator (fine-tuned LLM) with two complementary reward signals: a realism reward that encourages distributional fidelity, and a causal reward that promotes structural consistency. We refer to this approach as **CRAFT** (Causal Reward Aligned Fine-Tuning). Our findings are as follows:

**F1.** A scalar causal reward substantially improves intervention-consistent generation in controlled settings, without exposing the generator to structural equations.

**F2.** Proxy rewards estimated from observational data recover much of the oracle-reward benefit in our benchmarks.

**F3.** Causal alignment preserves distributional realism and actively improves manifold coverage.

From the structured-foundational model perspective, our results highlight a limitation of evaluating tabular generators only through distributional fidelity or downstream prediction. A model can be realistic and useful for Train-on-Synthetic, Test-on-Real (TSTR)-style prediction while failing under conditional intervention queries. We therefore argue that counterfactual queryability should be part of the reliability evaluation suite for structured-data foundation models. Notably, Tugnoli et al. (2026) find TabPFNGen (Ma et al., 2024a) strongly associates feature order with causal ordering, and propose permuting features to match the underlying causal graph; CRAFT instead induces causal coherence through implicit reward signals, removing the need for graph access. Although we instantiate CRAFT on an LLM-based queryable backbone, the alignment recipe is backbone-agnostic and applies to any queryable tabular foundation model whose outputs can be sampled and

scored.

## 2. Methodology

In this section, we define the scenario generation and counterfactual querying setting, and introduce the CRAFT methodology for enforcing causal alignment and realism through implicit reward signals. An overview of our proposed framework can be found in Figure 1.

**Problem Definition**   We study tabular data generation for two complementary use cases: **scenario generation**, and **counterfactual outcome querying**. In scenario generation, we sample a tuple $(X, T, Y)$, $X$ representing covariates, $T$ interventions, and $Y$ an outcome. For counterfactual prediction, we sample $Y$ conditional on $X$ and $T$. A useful generator must therefore satisfy two qualities: *realism* in maintaining the distributional properties of the original dataset, and *causal coherence*, by preserving the effects of interventions on downstream variables. Full details of methodology are provided in Appendix B.

**Baseline Pretrained Generator**   We replicate the fine-tuning method proposed by GREAT (Borisov et al., 2023), which adapts autoregressive large language models for tabular data generation. We initialize from a realism-trained checkpoint obtained by supervised fine-tuning Qwen3-1.7B (Yang et al., 2025) on the target dataset using next-token maximum likelihood estimation. This initialization provides a strong generator prior before reward-based fine-tuning.

**Realism Critic**   We use a Random Forest classifier as the realism critic or discriminator, initialized on a balanced dataset of real and generated rows, with the generated rows sampled from the pretrained baseline. It is periodically refit on a mixture of real rows, recently generated rows, and rows sampled from a replay buffer of earlier generator outputs. The replay buffer mitigates discriminator non-stationarity and stabilizes the realism reward across updates. For a generated row $x$, the discriminator reward is given by $r_d(x) = P_{\text{RF}}(\text{real} \mid x) \in [0, 1]$, the probability with which the discriminator believes the row to be real versus synthetic.

**Causal Evaluator**   The causal reward measures how faithfully the generated outcome $Y_{\text{gen}}$ is consistent with the causal mechanism linking covariates $X$ and treatment $T$ with the outcome. We distinguish two types of causal reward: oracle, where the true potential outcomes can be computed from the known data generating parameters, and proxy, where they are approximated from input data using a T-learner. We let $Y^\star(x, t)$ denote the expected outcome of a sample with covariates $x$ under treatment $t$, where $Y^\star(x, t)$ is computed from the structural model, or approximated

*Table 2.* Cov. denotes manifold coverage, Disc. denotes discriminator AUC, FE dentoes factual MAE, CF denotes counterfactual MAE, PEHE denotes $\sqrt{\text{PEHE}}$, and Sign denotes treatment-effect sign agreement. CRAFT-O is not reported on Twins because the data-generating parameters are unavailable in the real-world setting.

| | | Linear-Noiseless | | | | | | Twins | | | | | |
|---|---|---|---|---|---|---|---|---|---|---|---|---|---|
| Class | Model | Cov.↑ | Disc.↓ | FE↓ | CF↓ | PEHE↓ | Sign↑ | Cov.↑ | Disc.↓ | FE↓ | CF↓ | PEHE↓ | Sign↑ |
| *Ours* | CRAFT-O | **0.81** | 0.72 | 1816 | **2,528** | **3,564** | **0.85** | – | – | – | – | – | – |
| | CRAFT-P | 0.78 | 0.74 | 1834 | 3,012 | 4,284 | 0.81 | 0.60 | 0.92 | **0.06** | **0.18** | 0.43 | 0.82 |
| *Queryable* | GReaT | 0.40 | 0.74 | 11368 | 11,009 | 13,566 | 0.59 | 0.58 | 0.89 | 0.25 | 0.27 | 0.52 | 0.73 |
| | Claude | 0.34 | 0.99 | 4061 | 5,259 | 7,108 | 0.64 | 0.45 | 1.00 | 0.17 | 0.20 | 0.45 | 0.80 |
| | PredLLM | 0.06 | 0.99 | 24325 | 60,729 | 10,356 | 0.46 | 0.03 | 1.00 | 0.47 | 0.27 | 0.39 | 0.36 |
| *Realism-only* | TVAE | 0.45 | 0.96 | 4802 | – | 12,061 | 0.27 | 0.13 | 1.00 | 0.06 | – | **0.31** | **0.91** |
| | TabDDPM | 0.60 | 0.66 | 1033 | – | 6,953 | 0.55 | **0.88** | **0.70** | 0.21 | – | 0.39 | 0.36 |
| | TabSyn | 0.63 | **0.61** | **589** | – | 6,495 | 0.58 | 0.84 | 0.82 | 0.23 | – | 0.39 | 0.34 |

through a proxy estimator. Then, the causal reward is given by:

$$r_c(x) = \frac{1}{1 + \left|Y_{\text{gen}} - Y^{\star}(x,t)\right|/\sigma},\qquad(1)$$

with $r_c \in (0,1]$. $\sigma$ is a sharpness parameter, which is annealed during training (see Appendix B.3) to provide dense rewards in early phases, and stricter alignment later on. We note that even in the oracle reward setting, where we assume access to the underlying data generating function, the generator never observes the structural equations directly; it only receives reward signals based on generated outcomes.

**Dual-Reward Definition**  We fine-tune the pretrained generator with PPO using a critic-free advantage estimator. At each step, a batch of rows is sampled and scored along two axes, realism ($r_d$) and causal coherence ($r_c$), combined as

$$r(x) = (1-\alpha)\, r_d(x) + \alpha\, r_c(x)\qquad(2)$$

where $\alpha \in [0,1]$ is a tunable hyperparameter that weighs the two objectives. In this implementation $\alpha$ is held fixed over training. $\pi_{\text{ref}}$ is then subtracted:

$$r_{\text{eff}}(x) = r(x) - \beta \cdot D_{\text{KL}}(\pi_\theta \,\|\, \pi_{\text{ref}}),\qquad(3)$$

with $\beta$ being an hyperparameter and the Kullback-Leibler (KL) divergence summed over output tokens. Advantages are obtained by mean-centering and normalizing $r_{\text{eff}}$ within the batch (Shao et al., 2024), eliminating the need for a learned value network. A full specification of the discriminator, causal evaluator, KL distance computation and PPO specification can be found in Appendix B.2.

## 3. Results

### 3.1. Benchmark Settings and Evaluation Axis

We evaluate across four settings of increasing difficulty, spanning fully controlled simulation to real observational

data. All the details are provided in Appendix C. In Settings 1-3, we use the the Adult Census (Becker and Kohavi, 1996) dataset to provide a realistic population backbone, and simulate a binary treatment and outcome under three types of functional definitions: Linear-Noiseless, Linear-Noisy, and Non-Linear-Noisy. Setting 4 is the Twins benchmark (Louizos et al., 2017), where each twin pair provides with observational factual and counterfactual outcomes. A full description of the dataset construction can be found in C.

We evaluate generated samples along three axes: realism, causal fidelity, and downstream utility. Realism is measured by marginal-shape similarity, pairwise-dependence, manifold coverage, and discriminator AUC. Causal fidelity is measured by factual error (Fact-MAE), counterfactual error (CF-MAE), $\sqrt{\text{PEHE}}$, and treatment-effect sign agreement, capturing outcome accuracy under observed and intervened treatments as well as individual effect recovery. Downstream utility is measured by the Machine Learning Efficiency (MLE) gap. Detailed definitions are provided in Appendix D.2.

We provide a summary of results in Table 2 on two of the four settings, with full results on all metrics, confidence range, and all settings in Appendix E.

### 3.2. Causal Fidelity

**Counterfactual query accuracy.**  CRAFT's strongest gains occur on direct factual and paired counterfactual outcome prediction. Compared with the GReaT initialization and other queryable LLM baselines, CRAFT produces outcomes that are more consistent with intervention-conditioned responses, especially in the Linear-Noiseless and Twins settings. The Nonlinear-Noisy setting is harder: CRAFT improves factual consistency and treatment-direction agreement, but treatment-effect magnitude remains challenging.

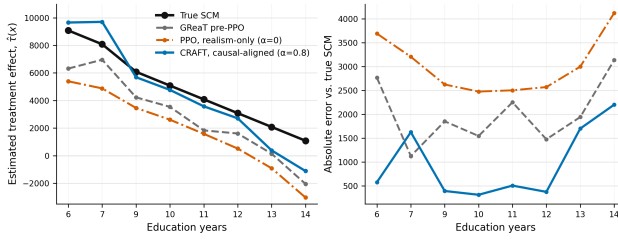

*Figure 2.* Estimated conditional treatment effects by education level in the Linear-Noiseless oracle setting. We plot the true SCM effect and model-estimated effects for GReaT, CRAFT $\alpha = 0$, and CRAFT $\alpha = 0.8$. Education buckets with fewer than 100 evaluation samples are excluded. The right panel shows absolute error relative to the true SCM effect (in $), highlighting that CRAFT $\alpha = 0.8$ tracks the causal mechanism more closely across most education levels.

| $\alpha$ | Cov.↑ | Fact.↓ | CF.↓ | PEHE↓ | Sign↑ |
|---|---|---|---|---|---|
| 0 | 0.44 | 8,941 | 9,462 | 11,764 | 0.57 |
| 0.8 | **0.81** | **1,816** | **2,528** | **3,564** | **0.85** |

*Figure 3.* Causal reward ablation in the Linear-Noiseless setting. Realism-only PPO improves little on counterfactual error, while causal-aligned PPO steadily reduces CF-MAE.

**Treatment-effect recovery.** Treatment-effect metrics provide a stricter test because they depend on accurately recovering both potential outcomes and their difference. CRAFT improves treatment-effect recovery in the simpler SCM and in Twins (measured by $\sqrt{\text{PEHE}}$), but its gains are not uniform in the nonlinear stochastic setting. This suggests that the causal reward is effective at aligning counterfactual responses, while fully recovering individual treatment-effect magnitudes under nonlinear noise remains difficult.

**Oracle vs. Proxy reward.** We observe that CRAFT trained on a data-estimated proxy reward maintains a similar performance to CRAFT trained oracle causal reward, where the true data-generating parameters are known, even outperforming the latter in certain metrics. This not only shows that a causal reward improves alignment, the same holds for causal reward computed in a setting where the underlying causal model is unknown.

Figure 3 isolates the effect of the causal reward. With $\alpha = 0$, PPO optimizes only realism and counterfactual error plateaus early. With $\alpha = 0.8$, CF-MAE steadily decreases

and treatment-effect recovery improves, indicating that the causal gains arise from reward alignment rather than RL fine-tuning alone.

### 3.3. Realism

**Distributional fidelity.** CRAFT largely preserves the observational fidelity of the GReaT initialization with marginal shapes and pairwise scores remaining comparable. As expected, distribution-only synthesizers such as TabDDPM and TabSyn remain strongest on several pure realism metrics, since they are optimized directly for joint-distribution matching (see Appendix E).

**Coverage.** Interestingly, CRAFT maintains higher coverage than all benchmark models in Settings 1-3, and has a higher coverage rate than queryable baselines in Setting 4. The $\alpha = 0$ ablation (Figure 4) suggests that this gain is not simply caused by PPO fine-tuning: realism-only PPO preserves surface-level fidelity, but does not produce the same coverage improvement. This supports the view that causal reward alignment encourages valid generation across a wider range of covariate-treatment combinations.

## 4. Conclusion

We show that causal structure can be induced in tabular generators without specifying or learning a structural causal model: a scalar causal reward, optimized over generated samples, steers a pretrained queryable generator toward intervention-consistent behavior, with gains on counterfactual accuracy, treatment-effect recovery, and rare-intervention reasoning even at fixed distributional realism. These gains persist when the reward is estimated from observational data, so the recipe is deployable where structural equations are unavailable.

This points to an alignment-first pathway for causal modeling in generative systems: rather than encoding the causal graph into the architecture, one can induce mechanism-consistent behavior through reward design. The recipe is backbone-agnostic and applies to any queryable tabular foundation model. Three limitations qualify these results: individual treatment-effect magnitudes under stochastic nonlinear mechanisms remain harder to recover than in the linear regime; we use a single 1.7B-parameter backbone and three of four settings are semi-synthetic; and we make no formal identifiability claim. Future work should scale reward-based alignment to larger backbones and richer modalities, evaluate on real interventional benchmarks beyond Twins, and characterize when implicit alignment recovers the true causal mechanism rather than approximating it within the observed support.

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

# A. Related Work

## A.1. Generative models

In this section, we distinguish between three classes of tabular data generators: realism-driven models, queryable generators, and causally explicit approaches. We compare these methods along three axes: (i) **queryability**, the ability to support conditional or counterfactual generation; (ii) **causal modeling**, whether the methods optimize for causal coherence; and (iii) **model-free generation**, the ability to operate without requiring explicit knowledge of the underlying causal model.

**Tabular Generators for Realism**  Drawing inspiration from advances in synthetic image generation, a first class of methods focuses on matching the distributional properties of tabular data. (Xu et al., 2019) introduce Conditional Tabular GANs (CT-GAN) and Tabular VAEs, which learn to generate samples by either fooling a discriminator or sampling from a learned latent space. More recent approaches leverage diffusion models: (Kotelnikov et al., 2023) learn to iteratively denoise random noise into realistic tabular rows, while (Zhang et al., 2023a) (TabSyn) combine diffusion with a VAE-based latent representation. These methods are effective at capturing marginal distributions and feature dependencies, but are not designed to model causal structure. As a result, they provide no guarantees on how generated samples respond to interventions. Moreover, conditional or partial row generation, which is required for counterfactual querying, is not natively supported, and typically requires ad hoc masking or imputation procedures. In addition, realism-driven generators can exhibit diversity-related failure modes: adversarial tabular models may suffer from mode collapse, VAE-based models from posterior collapse, and maximum-likelihood autoregressive models can concentrate probability mass on high-likelihood regions rather than preserving broad support over the data distribution (Xu et al., 2019; Li et al., 2025; Holtzman et al., 2020).

**Queryable Tabular Generators**  We refer to tabular generators that natively support conditional or partial row generation as *queryable* models. These systems allow users to specify a subset of covariates and generate the remaining features, making them naturally suited for counterfactual reasoning tasks. Large language models (LLMs) provide a flexible foundation for such generators. A simple approach consists of directly querying pretrained models (e.g., Claude (Anthropic, 2026), Pixtral (Mistral AI, 2024), Llama (Meta, 2024), (Meta AI, 2025)) to complete partially specified rows. More structured methods fine-tune LLMs for tabular generation: (Borisov et al., 2023) (GReaT) serialize tabular rows into token sequences and train the model using maximum likelihood with LoRA adapters, while (Nguyen et al., 2024a) (PredLLM) improve this framework through feature permu-

tation and enhanced training strategies to boost downstream utility. (Ling et al., 2024) build on GreAT by adversarially optimizing prompts using a realism discriminator; in our experiments, we find that discriminator reward is insufficient to enforce causal alignment. By construction, these methods support conditional data generation and therefore extend naturally to counterfactual querying. However, since they are trained to match observational distributions rather than interventional behavior, they provide no guarantees on the validity of counterfactual responses.

**Causally-Explicit Tabular Data Generators**  The third stream of tabular data generators encode causality directly in model design. Causal Generative Neural Networks (CGNN) learn functionals along edges of a structural causal model (SCM), and generate rows through conditional sampling (Goudet et al., 2017). Similarly, Causal-TGAN, relies on domain knowledge of the underlying causal graph to conditionally generate variables based on the values of the parent nodes (Wen et al., 2022). The Causal-Aware GAN method begins by learning the underlying DAG, and tailoring model architecture to sequentially sample downstream variables after parent nodes, with reinforcement learning to enforce alignment(Nguyen et al., 2025). While this method uses a causal reward, it also enforces a strong architectural prior, by building sequential subnetworks that match the underlying DAG, which is rarely available in real-world settings. These methods faithfully approach causally-grounded data generation, however they sidestep one of the key challenges in causal reasoning, which is lack of knowledge of the true causal model. In our work, we seek to show that causal parameters can be internalized without explicit modeling of the underlying system.

## A.2. Foundation Models for Tabular Data

A complementary line of work develops foundation models for tabular prediction. TabPFN (Hollmann et al., 2022) is a prior-fitted transformer trained on synthetic datasets that performs in-context learning on unseen tasks given a small number of demonstration rows. Ma et al. (2024a) extend this to full-row generation (TabPFNGen) by exploiting the autoregressive structure of the prior. Tugnoli et al. (2026) subsequently observe that TabPFNGen samples inherit a causal ordering from the sequence in which columns are presented, and propose to permute features in topological order along the underlying directed acyclic graph in order to recover causally-consistent generation. CRAFT is complementary: it offers a backbone-agnostic, reward-based alignment recipe that induces causal coherence in any queryable tabular foundation model — including TabPFNGen — without requiring access to a DAG or any structural reordering of features.

### A.3. Alignment for Generative Models

Post-training alignment has emerged as a powerful paradigm to adapt pre-trained generative models to objectives not captured by likelihood maximization training (Kumar et al., 2025). For LLMs, reinforcement learning methods such as Proximal Policy Optimization (PPO) (Ouyang et al., 2022) and Group Relative Policy Optimization (GRPO) (Shao et al., 2024) arise as methods for training a model to optimize non-differentiable or externally-validated rewards. In this work, we rely on PPO to fine-tune pre-trained baselines on rewards that capture a dual objective, capturing realism and causal coherence. Full finetuning of model weights is computationally very expensive, thus we use LoRA (Hu et al., 2021), a parameter-efficient alignment method, whereby only a small set of low-rank matrices is updated during training.

## B. Implementation and Experimental Setup

We build on GREAT (Borisov et al., 2023), a framework that fine-tunes a pre-trained large language model to generate synthetic tabular rows, using supervised fine-tuning. Each row is serialised as a comma-separated sequence of `column is value` pairs and generated autoregressively until the end-of-sequence token is produced. During training, columns are permuted so that there is no structural prior implicitly encoded by the column order (for example, covariates always preceding treatment and outcome). We use a GREAT checkpoint pre-trained on the target dataset with Qwen3-1.7B (Yang et al., 2025) as the backbone, and apply a fresh LoRA adapter (Hu et al., 2021) ($r = 16, \alpha_{\text{LoRA}} = 32$, dropout 0.1) on top of the merged weights. This adapter is the only component updated during PPO training; the pre-trained weights are frozen throughout. Table 3 shows a sample of generated rows. after training.

### B.1. Reinforcement Learning Fine-Tuning

We fine-tune the model with Proximal Policy Optimization (Schulman et al., 2017), following the RLHF paradigm (Ouyang et al., 2022) in which a frozen reference model $\pi_{\text{ref}}$ (in our case the GReaT finetuned checkpoint) provides a KL anchor. At each step, the policy $\pi_\theta$ samples a batch of $B = 16$ rows at temperature 0.9. Rows that fail to parse (missing columns, malformed values) receive a fixed penalty reward of $-0.5$; valid rows are scored by the reward function described below.

The clipped PPO surrogate loss is:

$$\mathcal{L}^{\text{PPO}}(\theta) = -\mathbb{E}\left[\min\left(\frac{\pi_\theta(a \mid s)}{\pi_{\text{old}}(a \mid s)}\hat{A},\right.\right.$$
$$\left.\left.\text{clip}\left(\frac{\pi_\theta(a \mid s)}{\pi_{\text{old}}(a \mid s)}, 1 - \varepsilon, 1 + \varepsilon\right)\hat{A}\right)\right]. \tag{4}$$

with clipping parameter $\varepsilon = 0.2$. Advantages $\hat{A}$ are computed from the effective per-row rewards (defined below) and normalized per batch, in GRPO style (Shao et al., 2024). This eliminates the need to train and run a separate value model. We run $K = 2$ gradient epochs per generated batch, with the Adam optimiser (learning rate $1.6 \times 10^{-5}$) and gradient clipping at $\ell_2$-norm 1.0.

### B.2. Reward Design

**Discriminator reward.** A Random Forest classifier (100 trees, max depth 5, min leaf size 4) is trained to distinguish real from generated rows, using a balanced mixture of 50% real rows, 25% current-batch synthetic rows, and 25% historical synthetic rows drawn from a circular replay buffer (capacity 500). The discriminator is retrained from scratch every 50 PPO steps and evaluated every 25 steps. For a generated row $x$, the discriminator reward is:

$$r_d(x) = P_{\text{RF}}(\text{real} \mid x) \in [0, 1]. \tag{5}$$

**Causal reward.** The causal reward measures how faithfully the generated outcome $Y_{\text{gen}}$ satisfies the structural equations given the covariates $X$ and treatment $T$. The reward is:

$$r_c(x) = \frac{1}{1 + \left|Y_{\text{gen}} - \hat{Y}(x, t)\right| / \sigma}, \tag{6}$$

with $r_c \in (0, 1]$. The sharpness parameter $\sigma$ is annealed during training (see below). In all settings, $\hat{Y}(x, t)$ is provided by a *proxy causal model*: a T-learner with XGBoost base regressors, trained once on 80% of the observed data and frozen throughout PPO, which fits separate outcome models $\mu_0(x)$ and $\mu_1(x)$ for the control and treated arms, giving $\hat{Y}(x, t) = \mu_t(x)$. In Settings 1 and 3, where the ground-truth SCM is available, we additionally evaluate the reward using the oracle potential deterministic outcome as $\hat{Y}(x, t)$.

**KL divergence computation.** The per-sequence KL penalty is computed as a sum of per-token log-probability ratios along the generated output. Concretely, let $x = (a_1, \ldots, a_T)$ be the sequence of tokens produced by the current policy. At each position $t$, both the current policy $\pi_\theta$ and the frozen reference policy $\pi_{\text{ref}}$ (the pre-trained GReaT checkpoint) assign a conditional distribution over the vocabulary given the preceding context. The KL term is then

| age | workclass | education | education-num | marital-status | occupation | relationship | race | sex | capital-gain | capital-loss | hours-per-week-work | native-country | treatment | outcome |
|---|---|---|---|---|---|---|---|---|---|---|---|---|---|---|
| 35 | Private | Some-college | 10 | Divorced | Adm-clerical | Not-in-family | White | Male | 0 | 0 | 40 | United-States | 0 | 97829.52 |
| 34 | State-gov | Bachelors | 13 | Married-civ-spouse | Exec-managerial | Husband | White | Male | 0 | 0 | 45 | United-States | 0 | 110203.18 |
| 26 | Private | HS-grad | 9 | Married-civ-spouse | Sales | Husband | Black | Male | 0 | 0 | 36 | United-States | 1 | 91314.14 |
| 23 | Private | Assoc-acdm | 12 | Never-married | Craft-repair | Not-in-family | White | Male | 0 | 0 | 40 | United-States | 1 | 98696.04 |
| 41 | State-gov | Bachelors | 13 | Divorced | Prof-specialty | Unmarried | White | Female | 0 | 0 | 40 | United-States | 0 | 114998.58 |

*Table 3.* Sample rows generated by our fine-tuned model in the linear noisy setting (Setting 2).

estimated via a single-sample Monte Carlo approximation using the tokens actually sampled:

$$D_{\mathrm{KL}}\big(\pi_\theta \,\|\, \pi_{\mathrm{ref}}\big) = \sum_{t=1}^{T} \log \frac{\pi_\theta(a_t \mid a_{<t})}{\pi_{\mathrm{ref}}(a_t \mid a_{<t})}. \qquad (7)$$

Because $a_t$ is drawn from $\pi_\theta$, Equation (7) is an unbiased estimator of the true KL divergence. Both log-probabilities are obtained from a single forward pass: $\pi_\theta$ is the live policy being optimized, while $\pi_{\mathrm{ref}}$ is kept frozen throughout training.

**Combined reward and KL penalty.**   The two signals are mixed via a scalar $\alpha \in [0, 1]$:

$$r(x) = (1 - \alpha)r_d(x) + \alpha r_c(x) \qquad (8)$$

To prevent the policy from drifting away from the pre-trained prior, we subtract a per-sequence KL penalty:

$$r_{\mathrm{eff}}(x) = r(x) - \beta \cdot D_{\mathrm{KL}}\big(\pi_\theta | \pi_{\mathrm{ref}}\big) \qquad (9)$$

where $\beta = 0.02$ controls the strength of the regularization, discouraging degenerate solutions in which the model maximizes reward at the cost of generating out-of-distribution text. Equation (9) is the signal that enters the PPO advantage computation.

**B.3. Sigma Curriculum**

The sharpness parameter $\sigma$ in Eq. (6) must be calibrated to the scale of the outcome variable: setting $\sigma = 1$ is uninformative when $Y$ takes values in the tens of thousands, and becomes particularly problematic under stochastic SCMs (Settings 2–4), where the irreducible noise floor $\sigma_Y$ already induces errors that would collapse the reward to near zero. A single fixed $\sigma$ is also insufficient across training: a large $\sigma$ makes the reward too flat to provide useful gradients once the model begins to improve, while a small $\sigma$ is too sparse at the start of training when errors are large. We therefore use a curriculum over $\sigma$.

Early in PPO training, generated outcomes can be far from the causal target, so a larger $\sigma$ gives a smoother reward signal and avoids immediate reward collapse. As training progresses, we reduce $\sigma$ so that the reward becomes stricter and increasingly distinguishes fine-grained causal errors.

We anneal $\sigma$ with a cosine schedule after a 50-step warmup:

$$\begin{aligned} \sigma(s) &= \sigma_{\mathrm{start}} + \frac{1 - \cos(\pi\rho(s))}{2}\left(\sigma_{\mathrm{end}} - \sigma_{\mathrm{start}}\right), \\ \rho(s) &= \frac{s - s_{\mathrm{warm}}}{S - s_{\mathrm{warm}}}. \end{aligned} \qquad (10)$$

where $s$ is the current PPO step, $S$ the total steps, $s_{\mathrm{warm}} = 50$. For the Adult Salary settings, outcomes are measured in dollars, so we use a fixed dollar-scale curriculum,

$$\sigma_{\mathrm{start}} = 15{,}000, \qquad \sigma_{\mathrm{end}} = 2{,}000.$$

This schedule was kept fixed across the Adult oracle and proxy reward variants because the outcome scale and the initial generator error scale are the same. The initial value provides a soft reward in the high-error regime, while the final value makes the reward sensitive to salary-level causal deviations; in the noisy Adult settings it is also on the order of the injected outcome noise.

For the Twins setting, the outcome is binary mortality, so the Adult dollar-scale calibration is inappropriate. We instead auto-calibrate $\sigma$ from the proxy reward model. We fit the proxy T-learner on $80\%$ of the real training data and compute factual residuals on a held-out $20\%$ calibration split:

$$e_i = y_i - \hat{m}_{t_i}(x_i), \qquad \hat{m}_t(x) \approx \mathbb{E}[Y \mid X = x, T = t].$$

We set

$$\sigma_{\mathrm{start}} = \mathrm{sd}(e_i).$$

When the curriculum is enabled, the endpoint is rescaled by preserving the configured start-to-end ratio. In our Twins runs this ratio is $0.10/0.75$, so

$$\sigma_{\mathrm{end}} = \frac{0.10}{0.75}\,\sigma_{\mathrm{start}}.$$

This makes the reward scale adaptive to the empirical uncertainty of the proxy model and to the Bernoulli outcome scale, while preserving the same relative sharpening over training.

We confirm that adding the sigma schedule to the reward mechanism drives most of the gain on the causal axis. As shown in Figure 4, ablating the $\sigma$ curriculum and relying on a fixed sharpness parameter causes a severe degradation in individual-level causal accuracy, with PEHE worsening by roughly 2.5x (from $3{,}564$ to $9{,}096$). Furthermore, without the smooth-to-strict reward transition provided by the schedule, the generator prematurely narrows its focus, causing manifold coverage to drop heavily from $0.813$ to $0.539$.

*Table 4.* Causal and utility effects of the $\sigma$ curriculum on Linear-Noiseless, oracle reward.

| $\sigma$ schedule | Causal | | | | Utility | | |
|---|---|---|---|---|---|---|---|
| | Fact-MAE ↓ | CF-MAE ↓ | $\sqrt{\text{PEHE}}$ ↓ | SignAgr ↑ | MLE-gap ↓ | TSTR-$\sqrt{\text{PEHE}}$ ↓ | TSTR-ATE ↓ |
| No $\sigma$ schedule | 2,027 | **2,241** | 9,096 | 0.374 | **0.084** | 7,279 | 4,463 |
| Full $\sigma$ schedule (CRAFT) | **1,816** | 2,528 | **3,564** | **0.852** | 0.105 | **6,396** | **3,544** |

*Table 5.* Realism metrics for the same ablation.

| $\sigma$ schedule | Shapes ↑ | Pairs ↑ | Cov. ↑ | Disc. AUC ↓ |
|---|---|---|---|---|
| No $\sigma$ schedule | **0.924** | 0.940 | 0.539 | **0.679** |
| Full $\sigma$ schedule (CRAFT) | 0.903 | **0.940** | **0.813** | 0.724 |

*Figure 4.* Ablation of the causal-reward $\sigma$ schedule on Linear-Noiseless, oracle reward. The curriculum substantially improves individual-level causal recovery and coverage, while preserving comparable marginal and pairwise realism.

*Table 6.* Hyperparameters used in all experiments unless otherwise noted.

| Component | Parameter | Value |
|---|---|---|
| Base model | Backbone | Qwen3-1.7B |
| | Max new tokens | 400 |
| | Sampling temp. | 0.9 |
| LoRA | Rank $r$ | 16 |
| | $\alpha_{\text{LoRA}}$ | 32 |
| | Dropout | 0.1 |
| PPO | Total steps $S$ | 1600 |
| | Batch size $B$ | 16 |
| | Grad. epochs $K$ | 2 |
| | Clip $\varepsilon$ | 0.2 |
| | Learning rate | $1.6 \times 10^{-5}$ |
| | Grad. clip | 1.0 |
| | KL coef. $\beta$ | 0.02 |
| | Fail reward | $-0.5$ |
| Discriminator | Trees | 100 |
| | Max depth | 5 |
| | Min leaf size | 4 |
| | Retrain every | 50 steps |
| | Replay buffer | 500 rows |
| Reward | $\alpha$ | 0.8 |
| | $\sigma_{\text{start}}$ Adult | 15,000 |
| | $\sigma_{\text{end}}$ Adult | 2,000 |
| | $\sigma_{\text{start}}$ Twins | sd$(Y - \hat{m}_T(X))$ on held-out calibration split |
| | $\sigma_{\text{end}}$ Twins | $(0.10/0.75)\sigma_{\text{start}}$ |

## B.4. Implementation Details

Table 6 summarises all hyperparameters.

All experiments were run on a single NVIDIA H100 GPU. Each GReaT pre-training run took approximately 2 hours, and each PPO fine-tuning run approximately 12 hours. Across the four settings, seeds, and baseline evaluations, the reported experiments totaled roughly 400 GPU-hours. Including preliminary experiments, hyperparameter exploration (e.g., $\alpha$, $\sigma$ schedule, KL coefficient), and discarded runs, total project compute was approximately 500 GPU-hours.

## C. Settings and Datasets

We evaluate our approach across four settings of increasing difficulty, each relaxing one or more idealizing assump-

tions from the previous. This progression allows us to isolate sources of difficulty and demonstrate robustness as we approach real-world conditions. Settings 1–3 are semi-synthetic, built on the Adult Census UCI dataset (Becker and Kohavi, 1996) (48,000 individuals, 15 variables including age, education, occupation, and salary); we drop the original salary column and simulate intervention assignments and post-intervention outcomes. Setting 4 uses a real causal dataset with no known ground-truth mechanism.

**Setting 1—Linear SCM, noiseless, oracle reward.** We begin with the simplest possible setting to validate the core mechanism. The structural causal model (SCM) is linear and deterministic, and we assume access to an *oracle reward*: the true individual treatment effect $\tau(X)$ is known and used directly to evaluate generated samples. We simulate a binary treatment $T \in \{0, 1\}$ and continuous outcome $Y \in \mathbb{R}_+$ via:

$$T \mid X \sim \text{Bernoulli}(\sigma(-1.5\,\texttt{age} - 0.5\,\texttt{edu} - 1)),$$
$$Y(X, T) = 30000 + 800\,\texttt{age} + 4000\,\texttt{edu} + \tau(X)\,T,$$
$$\tau(X) = 5000 - 1000\,\texttt{edu},$$

where $\texttt{edu}$ denotes $\texttt{education\_num}$. Because $\tau(X)$ is closed-form and noise-free, the reward signal is exact.

**Setting 2 — Linear SCM, Gaussian noise, proxy reward.** This setting addresses two weaknesses of Setting 1. First, real data-generating processes are stochastic; we introduce additive Gaussian noise on both the heterogeneous effect and the outcome. Second, oracle rewards are unavailable in practice; we replace the true $\tau(X)$ with a *proxy* estimated from observed data via a standard CATE estimator. The structural equations become:

$$Y(X, T) = 30000 + 800\,\texttt{age} + 4000\,\texttt{edu} + \tau(X)\,T + \varepsilon_Y,$$
$$\varepsilon_Y \sim \mathcal{N}(0, \sigma_Y^2),$$
$$\tau(X) = 5000 - 1000\,\texttt{edu} + \varepsilon_\tau,$$
$$\varepsilon_\tau \sim \mathcal{N}(0, \sigma_\tau^2).$$

This tests whether the method remains effective when the reward is noisy and estimated rather than exact. We set $\sigma_\tau = 2,000$.

**Setting 3 — Non-linear SCM.** We further relax the linearity assumption, replacing the additive-linear structural equations with non-linear counterparts that introduce multiplicative interactions and saturation effects. The heterogeneous treatment effect becomes:

$$\tau(X) = 5000 \cdot \sigma\,(\texttt{age} - 35) \cdot (1 + 0.3,\texttt{edu}) + \varepsilon_\tau,$$

with the noisy outcome model from Setting 2 otherwise unchanged.

**Setting 4 — Twins (real-world, unknown SCM).** Finally, we evaluate on the Twins dataset (Louizos et al., 2017), a real world causal benchmark derived from US twin birth records. Both twins are observed under distinct conditions, providing near-counterfactual outcomes without requiring experimental design. Unlike Settings 1–3, there is no known structural equation or ground-truth $\tau(X)$: this is the hardest evaluation, where correctness can only be assessed through downstream inference metrics and held-out factual outcomes.

**Inducing confounding.** The raw Twins dataset is essentially randomized: the assignment of which twin is observed is approximately independent of the covariates, so a model trained on it as-is would face an RCT rather than the selection-biased observational setting that motivates causal inference. Following the standard recipe used by CE-VAE (Louizos et al., 2017) and GANITE (Yoon et al., 2018), we re-sample $T$ from a propensity that depends on a hidden confounder. We restrict to low-birth-weight pairs (both twins below 2000g) and use `gestation_weeks_group` as the confounder, since it is correlated with infant mortality and is therefore not adjusted for by treatment alone. Concretely, let $z(x)$ be the standardized gestation-weeks group; we draw

$$\ell(x) = \beta\, z(x) + \varepsilon, \qquad \varepsilon \sim \mathcal{N}(0,\, \sigma_\ell^2), \qquad (11)$$

$$e(x) = \mathrm{clip}\big(\sigma(\ell(x)),\, 0.05,\, 0.95\big), \qquad (12)$$

$$T \mid X \sim \mathrm{Bernoulli}\big(e(X)\big), \qquad (13)$$

with $\beta = 1.5$ and $\sigma_\ell = 0.5$. The clipping enforces overlap and ensures every unit retains nonzero probability of either treatment. The factual outcome is then $Y = T \cdot \mathtt{mort\_1} + (1 - T) \cdot \mathtt{mort\_0}$ and the paired counterfactual is its complement, so the oracle ITE $\tau_i = \mathtt{mort\_1}_i - \mathtt{mort\_0}_i$ is preserved exactly while the observational distribution becomes genuinely confounded.

# D. Evaluation

## D.1. Models Evaluated

We evaluate our models against two categories of models: *generative models* that produce synthetic tabular data, and *CATE estimators*.

### D.1.1. GENERATIVE MODELS

**GReaT (Borisov et al., 2023).** GReaT serializes tabular rows as natural language sentences and fine-tunes a pretrained autoregressive language model (GPT-2) on them. Rows are generated autoregressively and parsed back into structured records. We use a dataset-specific GReaT fine-tune as both a standalone baseline and as the initialization for our PPO stage.

**CTGAN (Xu et al., 2019).** CTGAN uses a conditional GAN with mode-specific normalization for multimodal continuous columns and a training-by-sampling procedure to mitigate class imbalance in categorical variables.

**TVAE (Xu et al., 2019).** TVAE applies the same mode-specific normalization within a variational autoencoder, serving as a non-adversarial counterpart to CTGAN from the same work.

**CTAB-GAN (Zhao et al., 2021).** CTAB-GAN augments CTGAN with an auxiliary classifier loss and an information-loss conditional vector, improving utility on downstream predictive tasks.

**CTAB-GAN+ (Zhao et al., 2022).** CTAB-GAN+ refines CTAB-GAN with revised architectural components and additional preprocessing transformations for skewed and long-tailed distributions.

**TabDDPM (Kotelnikov et al., 2023).** TabDDPM applies denoising diffusion probabilistic models to tabular data, using Gaussian diffusion for continuous columns and multinomial diffusion for categorical ones.

**TabSyn (Zhang et al., 2023a).** TabSyn first encodes tabular rows into a continuous latent space via a VAE, then trains a score-based diffusion model in that latent space. Decoupling representation learning from generation avoids applying diffusion directly to mixed-type features.

**TapTap (Zhang et al., 2023b).** TapTap conditions a pretrained language model on a small set of real in-context examples to generate tabular rows without gradient updates at inference time.

**PredLLM (Nguyen et al., 2024b).** PredLLM fine-tunes a pretrained LLM on serialized tabular data using a novel input permutation strategy, and generates synthetic rows through a feature-conditional sampling procedure designed to accurately capture ground-truth feature-target correlations.

**CLLM (Seedat et al., 2024).** CLLM leverages the in-context learning capabilities of pretrained LLMs to generate synthetic tabular rows without fine-tuning, followed by a principled data curation phase that filters the generated samples based on aleatoric uncertainty and predictive confidence to maximize downstream utility in low-data regimes.

**Claude Opus 4.7 (Anthropic, 2026).** We prompt Claude Opus 4.7 zero-shot with the column schema and a fixed number of real rows as demonstrations, instructing it to

generate synthetic rows in the same format. No fine-tuning is performed.

**Llama 4 Maverick (Meta AI, 2025).** Prompted under the same zero-shot protocol as Claude Opus 4.7.

**Llama 3.3 70B (Meta, 2024).** Prompted under the same zero-shot protocol.

**Mistral Pixtral Large (Mistral AI, 2024).** Prompted under the same zero-shot protocol.

### D.1.2. CATE ESTIMATORS

The following estimators are trained directly on the real training data and evaluated on the real held-out test set against oracle $\tau_i$. As dedicated causal inference methods, they do not generate synthetic data, so they are not directly comparable to CRAFT, but they represent a practical ceiling on CATE estimation quality and serve as an upper-bound reference for the TSTR results.

**T-learner (Künzel et al., 2019).** Fits two separate outcome models $\hat{\mu}_0(x) = \mathbb{E}[Y \mid X{=}x, T{=}0]$ and $\hat{\mu}_1(x) = \mathbb{E}[Y \mid X{=}x, T{=}1]$ using gradient-boosted trees. The CATE estimate is $\hat{\tau}(x) = \hat{\mu}_1(x) - \hat{\mu}_0(x)$.

**DR-learner (Kennedy, 2023).** Constructs doubly robust pseudo-outcomes via cross-fitted propensity and outcome models,

$$\tilde{\tau}_i = \frac{t_i - \hat{e}(x_i)}{\hat{e}(x_i)(1 - \hat{e}(x_i))}\big(y_i - \hat{\mu}_{t_i}(x_i)\big) + \hat{\mu}_1(x_i) - \hat{\mu}_0(x_i),$$

then regresses $\tilde{\tau}_i$ on $x_i$ to obtain $\hat{\tau}(x)$.

**CausalPFN (Balazadeh et al., 2025).** CausalPFN is a transformer meta-learned on synthetic datasets drawn from a prior over structural causal models, enabling in-context CATE estimation at test time without task-specific training.

**DiffPO (Ma et al., 2024b).** DiffPO models the joint potential outcome distribution $(Y(0), Y(1))$ with a conditional diffusion model. CATE estimates are obtained as $\hat{\tau}(x) = \mathbb{E}[Y(1) - Y(0) \mid X{=}x]$ by marginalizing over samples from the diffusion chain.

### D.2. Evaluation Metrics

We describe each metric used across the three evaluation axes. Let $\mathcal{D}_{\text{real}} = \{(x_i, t_i, y_i)\}_{i=1}^{N}$ denote the real held-out test set and $\mathcal{D}_{\text{synth}} = \{(\tilde{x}_j, \tilde{t}_j, \tilde{y}_j)\}_{j=1}^{M}$ the synthetic samples. When oracle individual treatment effects are available, we write $\tau_i = Y_i(1) - Y_i(0)$ and $\hat{\tau}_i$ for the corresponding estimate derived from synthetic data.

### D.2.1. REALISM

**Column Shapes.** For each continuous column $k$ we compute the two-sample Kolmogorov–Smirnov statistic $\text{KS}_k$ between real and synthetic marginals. For each categorical column $k$ we compute the Total Variation Distance $\text{TVD}_k = \frac{1}{2}\sum_v |p_k(v) - \hat{p}_k(v)|$. The overall score is the mean across all columns:

$$\texttt{col\_shapes} = \frac{1}{K}\sum_{k=1}^{K} s_k, \qquad s_k = \begin{cases} \text{KS}_k & \text{continuous} \\ \text{TVD}_k & \text{categorical}. \end{cases}$$

Higher values indicate better marginal fidelity.

**Column-Pair Trends.** For each ordered pair of columns $(j, k)$ we compute a pairwise association score $a_{jk}$: Pearson correlation for two continuous columns, Cramér's V for two categorical columns, and the correlation between one column and the target-encoded version of the other for mixed pairs. The metric averages these scores over all $\binom{K}{2}$ pairs.

**Classifier Two-Sample Test (Disc-AUC).** A Random Forest classifier (200 trees, 3-fold CV) is trained to distinguish real from synthetic rows. We report the mean held-out AUC. A value of 0.5 indicates the two distributions are indistinguishable; values approaching 1.0 indicate the synthetic data is easily identified.

**Parse Rate.** Applicable to text-output generators (e.g. GReaT, LLM-based samplers). We report the fraction of raw generation attempts that produce a complete, schema-conforming row after parsing. A parse rate below 1.0 indicates the model occasionally produces malformed outputs.

**Coverage.** As proposed in (Naeem et al., 2020), coverage measures the fraction of real samples whose neighbourhoods contain at least one fake sample. For each real test point $r_i$, compute $R_i$ = distance to its $k$-th nearest real neighbor ($k = 5$). Then find the nearest synthetic point to $r_i$. A real point is "covered" if that nearest-synth distance $\leq R_i$.

$$\text{Coverage} = \frac{|i : \min_j |r_i - s_j| \leq R_i|}{N_{\text{real}}} \tag{14}$$

### D.2.2. CAUSAL FIDELITY

**Factual MAE.** For each synthetic row $(\tilde{x}, \tilde{t}, \tilde{y})$ we compute the absolute deviation from the oracle conditional mean outcome:

$$\texttt{Fact-MAE} = \frac{1}{M}\sum_{j=1}^{M}\big|\tilde{y}_j - \mathbb{E}[Y \mid X{=}\tilde{x}_j,\, T{=}\tilde{t}_j]\big|.$$

This measures whether the generator places outcome values at the correct location for each covariate–treatment combination, independently of treatment effect estimation.

**Paired Counterfactual MAE.** Given a real row $(x_i, t_i, y_i)$ with oracle counterfactual $y_i^{(1-t_i)}$, we retrieve the synthetic row generated under the flipped treatment and compute:

$$\text{CF-MAE} = \frac{1}{N}\sum_{i=1}^{N}\left|\hat{y}_i^{(1-t_i)} - y_i^{(1-t_i)}\right|.$$

This is the most direct measure of counterfactual accuracy and requires oracle potential outcomes.

**$\sqrt{\text{PEHE}}$ (Precision in Estimating Heterogeneous Effects).**

$$\sqrt{\text{PEHE}} = \sqrt{\frac{1}{N}\sum_{i=1}^{N}(\hat{\tau}_i - \tau_i)^2}.$$

This is the RMSE between the estimated and oracle individual treatment effects and is the primary individual-level causal metric.

**ATE Error.**

$$\text{ate\_error} = \left|\frac{1}{N}\sum_{i=1}^{N}\hat{\tau}_i - \frac{1}{N}\sum_{i=1}^{N}\tau_i\right|.$$

This measures population-level bias in the estimated average treatment effect, independently of individual-level accuracy.

**$\tau$-Wasserstein Distance.** The Wasserstein-1 distance between the distribution of estimated effects $\{\hat{\tau}_i\}$ and oracle effects $\{\tau_i\}$:

$$W_1(\hat{\tau}, \tau) = \inf_{\gamma \in \Pi(\hat{\tau}, \tau)} \mathbb{E}_{(u,v)\sim\gamma}[|u - v|].$$

Unlike ATE error, this penalises distributional mismatch even when the means coincide (e.g. mean-collapse).

**Sign Agreement.**

$$\text{sign\_agreement} = \frac{1}{N}\sum_{i=1}^{N}\mathbf{1}\big[\text{sign}(\hat{\tau}_i) = \text{sign}(\tau_i)\big].$$

This measures directional fidelity: whether the model correctly identifies which units benefit from treatment.

#### D.2.3. PAIRED POTENTIAL OUTCOMES PROTOCOL

All causal metrics that involve treatment effects require both potential outcomes $Y(x, T=0)$ and $Y(x, T=1)$ for each evaluation row. We obtain these via three pathways depending on the generator class:

**(a) Queryable generators** (Ours, GReaT, TapTap, PredLLM, third-party LLMs). We sample $Y$ conditional on $(X, T)$ separately under each treatment arm. This gives two outputs per evaluation row: the factual outcome $\hat{Y}(x, T=t)$ for the observed $t$, and the paired counterfactual $\hat{Y}(x, T=1-t)$.

**(b) Joint-only generators** (CTGAN, TVAE, CTAB-GAN/+, TabDDPM, TabSyn). These models only emit unconditional samples $(X, T, Y)$. To recover per-arm outcomes we apply $K=5$ nearest-neighbour matching across the synthetic pool: covariates are encoded with StandardScaler for numerics and OneHotEncoder for categoricals, with the transformer fit on $\mathcal{D}_{\text{real-test}} \cup \mathcal{D}_{\text{syn}}$. For each real test row $x$, we compute

$$\hat{\tau}(x) = \frac{1}{K}\sum_{j\in\mathcal{N}_K^{T=1}(x)} Y_j^{\text{syn}} - \frac{1}{K}\sum_{j\in\mathcal{N}_K^{T=0}(x)} Y_j^{\text{syn}}, \quad (15)$$

where $\mathcal{N}_K^{T=t}(x)$ are the $K$ nearest synthetic neighbours of $x$ restricted to treatment arm $t$. The matching is computed once per cell and shared across all metrics that consume $\hat{\tau}$.

**(c) CATE-only baselines** (T-learner, DR-learner, CausalPFN, DiffPO). These predict $\hat{\tau}(x)$ directly and we use it as-is. They do not emit joint samples, so realism and Fact-MAE are skipped for them.

**Reference outcomes $Y^\star(x, t)$.**

- **Settings 1–3 (Adult Salary semi-synthetic):** $Y^\star(x, t)$ is computed in closed form from the structural equations of the SCM (Appendix C).

- **Setting 4 (Twins):** the dataset directly supplies both potential outcomes per pair — mortality of the heavier twin ($Y^\star(x, 1)$) and of the lighter twin ($Y^\star(x, 0)$). $T$ is re-sampled from a propensity model on gestational age to induce confounding while preserving the paired counterfactual ground truth (Sec. C).

#### D.2.4. DOWNSTREAM UTILITY

**ML Efficiency (TSTR).** To evaluate the ML efficiency, we utilize the standard Train-on-Synthetic, Test-on-Real (TSTR) paradigm introduced by Esteban et al. (2017) Specifically, following the CTAB-GAN+ protocol (Zhao et al., 2021), we train a suite of four learners — Decision Tree, Linear Model, Random Forest, and Gradient Boosting — on $\mathcal{D}_{\text{synth}}$ and evaluate each on $\mathcal{D}_{\text{real}}$. We also train each learner on $\mathcal{D}_{\text{real}}$ (TRTR baseline). ML Efficiency is the mean absolute performance gap:

$$\text{ml\_efficiency} = \frac{1}{|\mathcal{L}|}\sum_{\ell\in\mathcal{L}}|\text{score}_\ell^{\text{TSTR}} - \text{score}_\ell^{\text{TRTR}}|,$$

where score is AUC for classification tasks and $R^2$ for regression. Lower values indicate that synthetic data supports downstream learning comparably to real data.

### D.3. Evaluation Protocol Details

**Dataset sizes.** Settings 1–3 use the UCI Adult Census dataset filtered to rows with valid covariates, yielding 32,564 rows total. Setting 4 uses the CEVAE-format Twins corpus filtered to low-birth-weight pairs (both twins below 2000g) with median imputation of sparse paternal/maternal covariates, yielding 11,006 same-sex twin pairs.

**Splits.** For every dataset and every seed we draw a single seeded permutation of the row (or pair) indices and partition it 70/15/15 into train/validation/test. From the same permutation we materialise two parallel views over the same rows: a *model* view containing only $(X, T, Y)$, and an *eval* view that additionally retains the oracle columns (`true_cate`, `counterfactual`, `propensity`, `logit`, when available). Concretely:

- **model-train + model-val**: GReaT pre-training, PPO fine-tuning, training of every generative baseline and every CATE estimator, and fitting of the proxy T-learner that supplies the causal reward in Settings 2–4. The few-shot LLM baselines (Claude Opus, Llama, Mistral) draw their in-context demonstrations from this split.
- **eval-test**: held-out set used for all reported metrics; the oracle columns provide ground-truth $\tau_i$ and counterfactuals $y_i^{(1-t_i)}$.

**Seeds and reported variability.** All numbers are averaged over $S = 4$ seeds. Each seed re-seeds (i) the dataset permutation, (ii) treatment sampling for the SCM, (iii) baseline initialisation and training, and (iv) the generation step, so the train/test partition rotates and the reported variance reflects both data-split and training stochasticity. Reported $\pm$ values are *standard deviations across the $S$ seeds* of the per-seed point estimates; For Twins, splitting is performed at the *pair* level so both potential outcomes for a pair always land in the same split. The $\sigma$-calibration procedure for Twins (App. B.3) further sub-divides the model-train split 80/20 into proxy-fit and residual-calibration partitions; the eval-test split is never touched by calibration.

**Preprocessing.** Each baseline uses its own native preprocessing pipeline as published, applied identically to model-train and to the rows used to score (model-train for fitting, eval-test for evaluation). Continuous columns are standardised and categoricals one-hot encoded for tree/MLP-based discriminator and CATE-estimator metrics; the encoders are fit jointly on $\mathcal{D}_{\text{real-test}} \cup \mathcal{D}_{\text{synth}}$ to keep the feature space aligned, and used identically across all evaluated models.

**Malformed generations.** Joint-only baselines (CTGAN, TVAE, CTAB-GAN/+, TabDDPM, TabSyn) emit structured tensors and have parse rate 1.0 by construction. Text-output generators (GReaT, our PPO model, TapTap, PredLLM, CLLM, Claude Opus, Llama, Mistral) parse each decoded string against the column schema; rows missing columns or with un-coercible values are discarded, and the sampler continues until $M$ valid rows are collected. The fraction of attempts that yielded a valid row is reported as the `Parse Rate` realism metric, so parse failures are not silently filtered: they surface explicitly as a lower parse rate. To prevent run-away loops on degenerate generators, the sampler aborts if fewer than $M/4$ rows are parsed after $4M$ attempts; in that case metrics that require $M$ rows are computed on whatever was collected and the parse rate flags the issue.

## E. Results

We report here the full results across the 4 settings, including all our models and evaluation suite. Values reported are averaged over four seeds, but for few instances were models were computationally expensive to run.

*Table 7.* Adult Census, linear, noiseless (Setting 1). Oracle/proxy PPO rows shown. Mean with standard deviation across four seeds is reported where available. Single-run baselines are shown without error bars.

| Model | Realism | | | | | Fact-MAE ↓ | Causal | | | | | Utility |
|---|---|---|---|---|---|---|---|---|---|---|---|---|
| | Shapes ↑ | Pairs ↑ | Parse ↑ | Cov ↑ | DISC AUC ↓ | | CF-MAE ↓ | $\sqrt{\text{PEHE}}$ ↓ | ATE-err ↓ | $\tau$-Wass ↓ | SignAgr ↑ | MLE-gap ↓ |
| **CRAFT (Oracle)** | $0.903_{\pm0.003}$ | $0.940_{\pm0.006}$ | 1.000 | $\mathbf{0.813}_{\pm0.173}$ | $0.724_{\pm0.013}$ | $1,816_{\pm53.1}$ | $\mathbf{2,528}_{\pm71.5}$ | $\mathbf{3,564}_{\pm31.9}$ | $587.3_{\pm95.5}$ | $\mathbf{1,827}_{\pm117.2}$ | $\mathbf{0.852}_{\pm0.003}$ | $0.105_{\pm0.024}$ |
| **CRAFT (Proxy)** | $0.893_{\pm0.003}$ | $0.928_{\pm0.011}$ | 1.000 | $0.775_{\pm0.164}$ | $0.743_{\pm0.010}$ | $1,834_{\pm123.7}$ | $3,012_{\pm68.9}$ | $4,284_{\pm194.9}$ | $1,352_{\pm87.8}$ | $2,393_{\pm109.2}$ | $0.812_{\pm0.011}$ | $0.161_{\pm0.077}$ |
| GReaT (pre-PPO) | $0.937_{\pm0.004}$ | $0.932_{\pm0.004}$ | 1.000 | $0.404_{\pm0.008}$ | $0.743_{\pm0.012}$ | $11,368_{\pm142.8}$ | $11,009_{\pm7.65}$ | $13,566_{\pm20.2}$ | $1,879_{\pm60.5}$ | $9,406_{\pm50.8}$ | $0.588_{\pm0.001}$ | $0.543_{\pm0.154}$ |
| TapTap | $0.908_{\pm0.002}$ | $0.913_{\pm0.003}$ | 1.000 | $0.184_{\pm0.007}$ | $0.916_{\pm0.007}$ | $14,225_{\pm293.5}$ | $109,456_{\pm5,636}$ | $9,114_{\pm1,059}$ | $1,806_{\pm1,306}$ | $5,426_{\pm799.5}$ | $0.632_{\pm0.063}$ | $1.676_{\pm0.119}$ |
| PredLLM | $0.863_{\pm0.010}$ | $0.905_{\pm0.013}$ | 1.000 | $0.055_{\pm0.021}$ | $0.989_{\pm0.005}$ | $24,325_{\pm1,372}$ | $60,729_{\pm4200.5}$ | $10,356_{\pm735.4}$ | $5,558_{\pm837.9}$ | $6,732_{\pm724.7}$ | $0.461_{\pm0.042}$ | $1.188_{\pm0.098}$ |
| CLLM | 0.860 | 0.865 | 1.000 | 0.249 | 0.959 | 5,847 | – | 10,978 | 8,530 | 8,587 | 0.277 | 0.457 |
| MALLM-GAN | 0.832 | 0.843 | 1.000 | 0.194 | 0.981 | 5,972 | – | 7,318 | 6,216 | 6,216 | 0.355 | 0.775 |
| Claude Opus 4.7 | 0.885 | 0.774 | 1.000 | 0.341 | 0.911 | 2,610 | 4,374 | 5,977 | 2,563 | 3,351 | 0.745 | 0.141 |
| Llama 4 Maverick | $0.870_{\pm0.012}$ | $0.807_{\pm0.013}$ | 1.000 | $0.247_{\pm0.002}$ | $0.960_{\pm0.035}$ | $5,338_{\pm1,327}$ | $7,047_{\pm605.4}$ | $9,199_{\pm900.8}$ | $4,509_{\pm381.1}$ | $5,973_{\pm407.4}$ | $0.546_{\pm0.017}$ | $0.448_{\pm0.013}$ |
| Llama 3.3 70B | $0.863_{\pm0.001}$ | $0.839_{\pm0.030}$ | 1.000 | $0.247_{\pm0.010}$ | $0.936_{\pm0.055}$ | $6,150_{\pm791.7}$ | $9,248_{\pm388.6}$ | $13,075_{\pm733.1}$ | $6,124_{\pm221.8}$ | $8,033_{\pm381.9}$ | $0.476_{\pm0.036}$ | $0.494_{\pm0.025}$ |
| Mistral Pixtral Large | $0.865_{\pm0.002}$ | $0.895_{\pm0.022}$ | $0.991_{\pm0.010}$ | $0.364_{\pm0.012}$ | $0.891_{\pm0.045}$ | $5,407_{\pm185.6}$ | $9,149_{\pm655.8}$ | $11,916_{\pm3,171}$ | $4,793_{\pm1,275}$ | $7,547_{\pm1,003}$ | $0.456_{\pm0.040}$ | $4.250_{\pm6.616}$ |
| CTGAN | $0.866_{\pm0.005}$ | $0.912_{\pm0.011}$ | 1.000 | $0.310_{\pm0.169}$ | $0.946_{\pm0.015}$ | $11,929_{\pm611.9}$ | – | $11,142_{\pm721.4}$ | $8,242_{\pm1,208}$ | $8,409_{\pm1,017}$ | $0.335_{\pm0.047}$ | $0.554_{\pm0.013}$ |
| TVAE | $0.865_{\pm0.014}$ | $0.921_{\pm0.021}$ | 1.000 | $0.464_{\pm0.161}$ | $0.954_{\pm0.005}$ | $4,728_{\pm102.7}$ | – | $11,050_{\pm2,259}$ | $9,066_{\pm2,433}$ | $9,066_{\pm2,433}$ | $0.311_{\pm0.106}$ | $27.872_{\pm22.365}$ |
| CTAB-GAN | $0.741_{\pm0.049}$ | $0.856_{\pm0.003}$ | 1.000 | $0.040_{\pm0.014}$ | 1.000 | $14,687_{\pm796.8}$ | – | $10,195_{\pm936.0}$ | $5,792_{\pm747.8}$ | $6,712_{\pm833.8}$ | $0.471_{\pm0.036}$ | $1.379_{\pm0.316}$ |
| CTAB-GAN+ | $0.736_{\pm0.047}$ | $0.857_{\pm0.002}$ | 1.000 | $0.037_{\pm0.016}$ | 1.000 | $14,313_{\pm907.2}$ | – | $9,407_{\pm1,472}$ | $4,296_{\pm2,572}$ | $5,896_{\pm1,470}$ | $0.554_{\pm0.155}$ | $1.375_{\pm0.310}$ |
| TabDDPM | $\mathbf{0.969}_{\pm0.004}$ | $0.935_{\pm0.011}$ | 1.000 | $0.601_{\pm0.197}$ | $0.667_{\pm0.040}$ | $1,070_{\pm87.8}$ | – | $6,927_{\pm850.2}$ | $4,999_{\pm906.6}$ | $5,020_{\pm891.6}$ | $0.552_{\pm0.078}$ | $0.035_{\pm0.029}$ |
| TabSyn | $0.967_{\pm0.006}$ | $\mathbf{0.962}_{\pm0.008}$ | 1.000 | $0.633_{\pm0.191}$ | $\mathbf{0.614}_{\pm0.031}$ | $589.4_{\pm83.2}$ | – | $6,495_{\pm733.2}$ | $4,657_{\pm868.7}$ | $4,669_{\pm858.0}$ | $0.580_{\pm0.067}$ | $\mathbf{0.015}_{\pm0.007}$ |
| *CATE estimators (point estimate of $\tau(x)$ only; narrower task — reference ceiling, not direct competitors)* | | | | | | | | | | | | |
| T-learner | – | – | – | – | – | – | $356.5_{\pm0.81}$ | $653.8_{\pm3.45}$ | $75.6_{\pm1.09}$ | $295.4_{\pm1.13}$ | $0.979_{\pm0.000}$ | – |
| DR-learner | – | – | – | – | – | – | $125.3_{\pm15.9}$ | $436.8_{\pm82.7}$ | $22.6_{\pm7.68}$ | $103.5_{\pm10.1}$ | $0.998_{\pm0.001}$ | – |
| CausalPFN | – | – | – | – | – | – | $391.6_{\pm1.69}$ | $592.9_{\pm10.2}$ | $127.9_{\pm14.7}$ | $274.2_{\pm2.79}$ | $0.987_{\pm0.002}$ | – |
| DiffPO | – | – | – | – | – | – | $346.2_{\pm31.0}$ | $500.9_{\pm26.4}$ | $90.5_{\pm78.5}$ | $302.6_{\pm13.7}$ | $0.994_{\pm0.004}$ | – |

*Table 8.* Adult Census, linear, noise (Setting 2). Oracle/proxy PPO rows shown. Mean with STD across four seeds is reported where available (for computationally expensive model we only report one run)

| Model | Realism | | | | | Fact-MAE ↓ | Causal | | | | | Utility |
|---|---|---|---|---|---|---|---|---|---|---|---|---|
| | Shapes ↑ | Pairs ↑ | Parse ↑ | Cov ↑ | DISC AUC ↓ | | CF-MAE ↓ | $\sqrt{\text{PEHE}}$ ↓ | ATE-err ↓ | $\tau$-Wass ↓ | SignAgr ↑ | MLE-gap ↓ |
| **CRAFT (Oracle)** | $0.897_{\pm0.006}$ | $0.932_{\pm0.012}$ | 1.000 | $0.791_{\pm0.150}$ | $0.759_{\pm0.006}$ | $\mathbf{1,066}_{\pm18.8}$ | $\mathbf{2,058}_{\pm44.8}$ | $\mathbf{2,648}_{\pm71.2}$ | $702.0_{\pm57.9}$ | $\mathbf{1,156}_{\pm57.0}$ | $0.887_{\pm0.008}$ | $0.052_{\pm0.007}$ |
| **CRAFT (Proxy)** | $0.936_{\pm0.002}$ | $0.947_{\pm0.009}$ | 1.000 | $\mathbf{0.823}_{\pm0.174}$ | $0.700_{\pm0.024}$ | $1,308_{\pm33.2}$ | $2,123_{\pm53.5}$ | $2,698_{\pm80.5}$ | $503.9_{\pm48.2}$ | $1,230_{\pm72.7}$ | $\mathbf{0.891}_{\pm0.012}$ | $298.227_{\pm596.382}$ |
| GReaT (pre-PPO) | $0.938_{\pm0.004}$ | $0.953_{\pm0.007}$ | 1.000 | $0.555_{\pm0.011}$ | $0.662_{\pm0.016}$ | $2,677_{\pm126.6}$ | $2,528_{\pm27.4}$ | $3,217_{\pm18.3}$ | $\mathbf{363.9}_{\pm23.3}$ | $1,514_{\pm16.6}$ | $0.876_{\pm0.001}$ | $0.055_{\pm0.009}$ |
| TapTap | $0.911_{\pm0.004}$ | $0.906_{\pm0.009}$ | 1.000 | $0.188_{\pm0.010}$ | $0.924_{\pm0.003}$ | $15,390_{\pm662.5}$ | $133,394_{\pm2,442}$ | $9,098_{\pm1,212}$ | $1,724_{\pm925.8}$ | $5,278_{\pm953.1}$ | $0.651_{\pm0.077}$ | $1.850_{\pm0.082}$ |
| PredLLM | $0.865_{\pm0.011}$ | $0.900_{\pm0.004}$ | 1.000 | $0.157_{\pm0.139}$ | $0.993_{\pm0.004}$ | $25,357_{\pm1,406}$ | $16,593_{\pm6,721}$ | $10,866_{\pm1,219}$ | $4,349_{\pm1,437}$ | $6,711_{\pm708.7}$ | $0.507_{\pm0.062}$ | $1.209_{\pm0.045}$ |
| CLLM | 0.879 | 0.792 | 1.000 | 0.225 | 0.964 | 10,727 | – | 17,252 | 14,382 | 14,592 | 0.203 | 0.479 |
| MALLM-GAN | 0.832 | 0.747 | 1.000 | 0.116 | 0.989 | 8,448 | – | 6,045 | 5,149 | 5,149 | 0.489 | 0.824 |
| Claude Opus 4.7 | 0.905 | 0.782 | 1.000 | 0.412 | 0.992 | 13,995 | 15,019 | 19,405 | 4,464 | 13,445 | 0.530 | 0.337 |
| Llama 4 Maverick | $0.881_{\pm0.001}$ | $0.837_{\pm0.002}$ | 1.000 | $0.252_{\pm0.087}$ | $0.980_{\pm0.006}$ | $7,630_{\pm1,631}$ | $11,175_{\pm3,959}$ | $14,921_{\pm4,429}$ | $7,163_{\pm564.4}$ | $10,079_{\pm3,787}$ | $0.497_{\pm0.031}$ | $0.473_{\pm0.135}$ |
| Llama 3.3 70B | $0.888_{\pm0.001}$ | $0.821_{\pm0.008}$ | 1.000 | $0.293_{\pm0.127}$ | $0.972_{\pm0.008}$ | $8,130_{\pm2,092}$ | $11,897_{\pm4,910}$ | $16,248_{\pm5,540}$ | $4,052_{\pm144.1}$ | $10,557_{\pm4,814}$ | $0.554_{\pm0.038}$ | $0.483_{\pm0.145}$ |
| Mistral Pixtral Large | $0.901_{\pm0.005}$ | $0.909_{\pm0.012}$ | $0.997_{\pm0.001}$ | $0.462_{\pm0.174}$ | $0.896_{\pm0.006}$ | $7,523_{\pm2,481}$ | $11,350_{\pm5,018}$ | $14,082_{\pm5,597}$ | $2,945_{\pm2,937}$ | $9,257_{\pm4,195}$ | $0.559_{\pm0.127}$ | |
| CTGAN | $0.857_{\pm0.014}$ | $0.914_{\pm0.013}$ | 1.000 | $0.337_{\pm0.229}$ | $0.952_{\pm0.023}$ | $13,647_{\pm2,592}$ | – | $12,544_{\pm2,211}$ | $8,638_{\pm1,189}$ | $9,112_{\pm1,597}$ | $0.364_{\pm0.018}$ | $0.487_{\pm0.158}$ |
| TVAE | $0.872_{\pm0.015}$ | $0.903_{\pm0.022}$ | 1.000 | $0.512_{\pm0.172}$ | $0.955_{\pm0.006}$ | $5,289_{\pm1,588}$ | – | $14,744_{\pm3,974}$ | $12,928_{\pm3,587}$ | $12,928_{\pm3,587}$ | $0.173_{\pm0.076}$ | $20.754_{\pm17.188}$ |
| CTAB-GAN | $0.753_{\pm0.041}$ | $0.853_{\pm0.001}$ | 1.000 | $0.045_{\pm0.019}$ | $1.000_{\pm0.000}$ | $15,001_{\pm1,988}$ | – | $9,624_{\pm1,537}$ | $5,643_{\pm2,328}$ | $6,372_{\pm1,914}$ | $0.473_{\pm0.122}$ | $1.533_{\pm0.207}$ |
| CTAB-GAN+ | $0.748_{\pm0.039}$ | $0.854_{\pm0.001}$ | 1.000 | $0.047_{\pm0.019}$ | $1.000_{\pm0.000}$ | $14,769_{\pm1,983}$ | – | $9,905_{\pm1,395}$ | $6,158_{\pm2,040}$ | $6,689_{\pm1,719}$ | $0.448_{\pm0.113}$ | $1.629_{\pm0.284}$ |
| TabDDPM | $0.967_{\pm0.006}$ | $0.922_{\pm0.022}$ | 1.000 | $0.588_{\pm0.214}$ | $0.675_{\pm0.036}$ | $4,907_{\pm5,444}$ | – | $8,143_{\pm2,126}$ | $4,794_{\pm373.3}$ | $5,594_{\pm1,240}$ | $0.553_{\pm0.023}$ | $0.039_{\pm0.010}$ |
| TabSyn | $\mathbf{0.969}_{\pm0.005}$ | $\mathbf{0.964}_{\pm0.006}$ | 1.000 | $0.618_{\pm0.197}$ | $\mathbf{0.622}_{\pm0.012}$ | $4,449_{\pm5,225}$ | – | $7,464_{\pm2,507}$ | $4,284_{\pm330.0}$ | $5,056_{\pm1,588}$ | $0.582_{\pm0.048}$ | $\mathbf{0.022}_{\pm0.007}$ |
| *CATE estimators (point estimate of $\tau(x)$ only; narrower task — reference ceiling, not direct competitors)* | | | | | | | | | | | | |
| T-learner | – | – | – | – | – | – | $4,261_{\pm5,192}$ | $1,259_{\pm975.4}$ | $140.3_{\pm114.7}$ | $556.3_{\pm475.6}$ | $0.949_{\pm0.044}$ | – |
| DR-learner | – | – | – | – | – | – | $234.5_{\pm15.9}$ | $660.2_{\pm109.3}$ | $9.17_{\pm6.58}$ | $176.2_{\pm10.8}$ | $0.986_{\pm0.002}$ | – |
| CausalPFN | – | – | – | – | – | – | $1,322_{\pm1,407}$ | $1,694_{\pm1,695}$ | $74.5_{\pm130.7}$ | $922.5_{\pm1,158}$ | $0.937_{\pm0.095}$ | – |
| DiffPO | – | – | – | – | – | – | $4,705_{\pm5,338}$ | $2,999_{\pm2,646}$ | $1,182_{\pm1,041}$ | $1,869_{\pm1,870}$ | $0.853_{\pm0.154}$ | – |

*Table 9.* Adult Census, nonlinear, noise (Setting 3). Oracle/proxy PPO rows shown. Mean with STD across four seeds is reported where available (for computationally expensive model we only report one run)

| Model | Realism | | | | | Causal | | | | | | Utility |
|---|---|---|---|---|---|---|---|---|---|---|---|---|
| | Shapes ↑ | Pairs ↑ | Parse ↑ | Cov ↑ | DISC AUC ↓ | Fact-MAE ↓ | CF-MAE ↓ | $\sqrt{\text{PEHE}}$ ↓ | ATE-err ↓ | $\tau$-Wass ↓ | SignAgr ↑ | MLE-gap ↓ |
| **CRAFT (Oracle)** | $0.868_{\pm0.003}$ | $0.932_{\pm0.003}$ | 1.000 | $0.720_{\pm0.169}$ | $0.813_{\pm0.008}$ | **776.0**$_{\pm15.8}$ | **2,344**$_{\pm187.3}$ | $5,739_{\pm2,826}$ | $411.0_{\pm160.3}$ | **1,425**$_{\pm129.5}$ | $0.722_{\pm0.010}$ | $0.078_{\pm0.050}$ |
| **CRAFT (Proxy)** | $0.876_{\pm0.004}$ | $0.930_{\pm0.002}$ | 1.000 | **0.741**$_{\pm0.161}$ | $0.800_{\pm0.012}$ | $865.5_{\pm12.7}$ | $2,535_{\pm187.9}$ | $6,880_{\pm2,517}$ | $655.9_{\pm184.6}$ | $1,646_{\pm190.5}$ | $0.722_{\pm0.008}$ | $0.047_{\pm0.064}$ |
| GReaT (pre-PPO) | $0.933_{\pm0.007}$ | $0.935_{\pm0.005}$ | 1.000 | $0.561_{\pm0.009}$ | $0.654_{\pm0.012}$ | $2,590_{\pm94.8}$ | $2,587_{\pm42.0}$ | **3,381**$_{\pm36.9}$ | $63.5_{\pm8.82}$ | $1,429_{\pm31.0}$ | $0.681_{\pm0.004}$ | $1.462_{\pm0.163}$ |
| TapTap | $0.897_{\pm0.005}$ | $0.929_{\pm0.002}$ | 1.000 | $0.188_{\pm0.009}$ | $0.928_{\pm0.005}$ | $49,629_{\pm2,439}$ | 91,226 | $31,225_{\pm2,665}$ | $6,059_{\pm4,567}$ | $23,467_{\pm1,670}$ | $0.456_{\pm0.067}$ | $1.465_{\pm0.066}$ |
| PredLLM | $0.861_{\pm0.006}$ | $0.910_{\pm0.010}$ | 1.000 | $0.071_{\pm0.054}$ | $0.997_{\pm0.002}$ | $51,083_{\pm994.8}$ | 32,611 | $31,574_{\pm3,419}$ | $7,409_{\pm6,422}$ | $23,296_{\pm3,119}$ | $0.592_{\pm0.102}$ | |
| CLLM | 0.866 | 0.828 | 1.000 | 0.261 | 0.946 | 15,001 | – | 25,651 | 13,910 | 15,033 | 0.824 | 0.403 |
| MALLM-GAN | 0.850 | 0.851 | 1.000 | 0.210 | 0.984 | 27,634 | – | 7,209 | 3,232 | 4,395 | 0.798 | 0.853 |
| Claude Opus 4.7 | 0.897 | 0.824 | 1.000 | 0.452 | 0.992 | 7,278 | 8,194 | 22,783 | 3,022 | 7,176 | 0.600 | 0.141 |
| Llama 4 Maverick | $0.875_{\pm0.003}$ | $0.834_{\pm0.003}$ | 1.000 | $0.301_{\pm0.081}$ | $0.981_{\pm0.006}$ | $16,169_{\pm417.3}$ | $17,760_{\pm407.3}$ | $29,012_{\pm964.2}$ | $9,996_{\pm428.0}$ | $16,843_{\pm383.1}$ | $0.664_{\pm0.020}$ | $0.318_{\pm0.026}$ |
| Llama 3.3 70B | $0.852_{\pm0.003}$ | $0.823_{\pm0.004}$ | 1.000 | $0.304_{\pm0.098}$ | $0.974_{\pm0.009}$ | $19,092_{\pm474.3}$ | $17,747_{\pm815.2}$ | $27,964_{\pm1,568}$ | $8,287_{\pm1,022}$ | $16,796_{\pm871.4}$ | $0.672_{\pm0.023}$ | $0.431_{\pm0.026}$ |
| Mistral Pixtral Large | $0.874_{\pm0.003}$ | $0.928_{\pm0.009}$ | $0.997_{\pm0.002}$ | $0.415_{\pm0.124}$ | $0.944_{\pm0.014}$ | $26,912_{\pm688.2}$ | $22,053_{\pm420.7}$ | $33,257_{\pm976.9}$ | $12,066_{\pm1,712}$ | $21,290_{\pm447.2}$ | $0.674_{\pm0.013}$ | $0.577_{\pm0.101}$ |
| CTGAN | $0.872_{\pm0.009}$ | $0.921_{\pm0.011}$ | 1.000 | $0.387_{\pm0.168}$ | $0.941_{\pm0.011}$ | $18,844_{\pm546.9}$ | – | $36,873_{\pm4,456}$ | $19,667_{\pm5,069}$ | $24,086_{\pm2,963}$ | $0.782_{\pm0.046}$ | $0.341_{\pm0.062}$ |
| TVAE | $0.871_{\pm0.016}$ | $0.924_{\pm0.009}$ | 1.000 | $0.534_{\pm0.135}$ | $0.944_{\pm0.005}$ | $7,911_{\pm1,169}$ | – | $42,029_{\pm9,491}$ | $26,215_{\pm8,158}$ | $27,308_{\pm7,512}$ | **0.901**$_{\pm0.046}$ | $28.105_{\pm39.138}$ |
| CTAB-GAN | $0.780_{\pm0.007}$ | $0.864_{\pm0.012}$ | 1.000 | $0.043_{\pm0.022}$ | 1.000 | $54,838_{\pm424.8}$ | – | $33,787_{\pm3,248}$ | $6,904_{\pm7,777}$ | $25,141_{\pm2,251}$ | $0.561_{\pm0.118}$ | $1.964_{\pm0.335}$ |
| CTAB-GAN+ | $0.757_{\pm0.042}$ | $0.863_{\pm0.011}$ | 1.000 | $0.031_{\pm0.018}$ | $1.000_{\pm0.000}$ | $55,432_{\pm1,293}$ | – | $32,412_{\pm1,946}$ | $10,148_{\pm8,954}$ | $24,594_{\pm1,724}$ | $0.620_{\pm0.149}$ | $1.729_{\pm0.348}$ |
| TabDDPM | **0.969**$_{\pm0.005}$ | $0.930_{\pm0.023}$ | 1.000 | $0.580_{\pm0.199}$ | $0.677_{\pm0.038}$ | $3,709_{\pm355.5}$ | – | $22,053_{\pm2,472}$ | $9,662_{\pm2,022}$ | $11,550_{\pm2,162}$ | $0.777_{\pm0.025}$ | $0.048_{\pm0.026}$ |
| TabSyn | $0.965_{\pm0.007}$ | **0.957**$_{\pm0.007}$ | 1.000 | $0.615_{\pm0.201}$ | **0.641**$_{\pm0.019}$ | $2,773_{\pm281.1}$ | – | $18,290_{\pm3,077}$ | $5,692_{\pm2,548}$ | $8,845_{\pm2,708}$ | $0.713_{\pm0.034}$ | $0.008_{\pm0.005}$ |
| *CATE estimators (point estimate of $\tau(x)$ only; narrower task — reference ceiling, not direct competitors)* | | | | | | | | | | | | |
| T-learner | – | – | – | – | – | – | $1,636_{\pm1.64}$ | $523.1_{\pm5.83}$ | $29.3_{\pm3.23}$ | $143.7_{\pm9.36}$ | $0.862_{\pm0.015}$ | – |
| DR-learner | – | – | – | – | – | – | $236.3_{\pm7.79}$ | $485.2_{\pm62.8}$ | $14.4_{\pm6.59}$ | $94.7_{\pm11.8}$ | $0.933_{\pm0.008}$ | – |
| CausalPFN | – | – | – | – | – | – | $1,472_{\pm83.9}$ | $3,910_{\pm352.4}$ | $417.9_{\pm51.1}$ | $801.9_{\pm85.3}$ | $0.808_{\pm0.016}$ | – |
| DiffPO | – | – | – | – | – | – | $2,469_{\pm494.7}$ | $3,133_{\pm917.8}$ | $665.3_{\pm346.3}$ | $1,391_{\pm934.0}$ | $0.684_{\pm0.097}$ | – |

*Table 10.* Twins Dataset, Setting 4. Only Proxy PPO is available by design. Mean with STD across four seeds is reported where available (for computationally expensive model we only report one run). Both CTAB-GAN and CTAB-GAN+ are not able to generate rows on this dataset.

| Model | Realism | | | | | Causal | | | | | | Utility |
|---|---|---|---|---|---|---|---|---|---|---|---|---|
| | Shapes ↑ | Pairs ↑ | Parse ↑ | Cov ↑ | DISC AUC ↓ | Fact-MAE ↓ | CF-MAE ↓ | $\sqrt{\text{PEHE}}$ ↓ | ATE-err ↓ | $\tau$-Wass ↓ | SignAgr ↑ | MLE-gap ↓ |
| **CRAFT (Proxy)** | $0.924_{\pm0.001}$ | $0.964_{\pm0.001}$ | $0.999_{\pm0.001}$ | $0.597_{\pm0.049}$ | $0.918_{\pm0.003}$ | **0.06**$_{\pm0.00}$ | $0.18_{\pm0.01}$ | $0.43_{\pm0.01}$ | $0.09_{\pm0.01}$ | $0.09_{\pm0.01}$ | $0.819_{\pm0.005}$ | $0.211_{\pm0.012}$ |
| GReaT (pre-PPO) | $0.939_{\pm0.002}$ | $0.964_{\pm0.001}$ | 1.000 | $0.575_{\pm0.007}$ | $0.894_{\pm0.003}$ | $0.25_{\pm0.01}$ | $0.27_{\pm0.00}$ | $0.52_{\pm0.00}$ | $0.13_{\pm0.00}$ | $0.18_{\pm0.00}$ | $0.733_{\pm0.003}$ | $0.200_{\pm0.010}$ |
| TapTap | $0.949_{\pm0.002}$ | $0.961_{\pm0.001}$ | 1.000 | $0.454_{\pm0.018}$ | $0.868_{\pm0.004}$ | $0.32_{\pm0.01}$ | 0.90 | $0.77_{\pm0.67}$ | $0.03_{\pm0.03}$ | $0.39_{\pm0.37}$ | $0.306_{\pm0.069}$ | $0.209_{\pm0.006}$ |
| PredLLM | $0.843_{\pm0.021}$ | $0.953_{\pm0.001}$ | 1.000 | $0.033_{\pm0.009}$ | $1.000_{\pm0.000}$ | $0.47_{\pm0.05}$ | 0.27 | $0.39_{\pm0.01}$ | $0.07_{\pm0.04}$ | $0.18_{\pm0.01}$ | $0.356_{\pm0.034}$ | $0.210_{\pm0.006}$ |
| CLLM | 0.913 | 0.957 | 0.915 | 0.513 | 0.947 | 0.39 | – | 0.43 | 0.11 | 0.23 | 0.269 | |
| MALLM-GAN | 0.895 | 0.953 | 0.900 | 0.465 | 0.992 | 0.36 | – | 0.43 | 0.15 | 0.23 | 0.260 | 0.189 |
| Claude Opus 4.7 | 0.918 | 0.866 | 0.858 | 0.450 | 0.998 | 0.17 | 0.20 | 0.45 | **0.02** | 0.13 | 0.800 | **0.073** |
| Llama 4 Maverick | $0.944_{\pm0.002}$ | $0.938_{\pm0.003}$ | $0.882_{\pm0.009}$ | $0.661_{\pm0.049}$ | $0.952_{\pm0.018}$ | $0.19_{\pm0.01}$ | $0.20_{\pm0.01}$ | $0.45_{\pm0.01}$ | $0.03_{\pm0.01}$ | $0.11_{\pm0.01}$ | $0.801_{\pm0.010}$ | $0.144_{\pm0.013}$ |
| Llama 3.3 70B | $0.914_{\pm0.001}$ | $0.955_{\pm0.001}$ | $0.913_{\pm0.002}$ | $0.533_{\pm0.015}$ | $0.952_{\pm0.003}$ | $0.36_{\pm0.01}$ | **0.17**$_{\pm0.018}$ | $0.42_{\pm0.01}$ | $0.06_{\pm0.01}$ | $0.10_{\pm0.011}$ | $0.825_{\pm0.021}$ | $0.176_{\pm0.019}$ |
| Mistral Pixtral Large | $0.934_{\pm0.001}$ | $0.957_{\pm0.003}$ | $0.872_{\pm0.004}$ | $0.673_{\pm0.017}$ | $0.901_{\pm0.005}$ | $0.33_{\pm0.01}$ | $0.26_{\pm0.02}$ | $0.50_{\pm0.02}$ | $0.11_{\pm0.01}$ | $0.19_{\pm0.03}$ | $0.745_{\pm0.016}$ | $0.187_{\pm0.016}$ |
| CTGAN | $0.941_{\pm0.007}$ | $0.950_{\pm0.000}$ | 1.000 | $0.271_{\pm0.022}$ | $0.995_{\pm0.002}$ | $0.29_{\pm0.03}$ | – | $0.38_{\pm0.03}$ | $0.03_{\pm0.02}$ | $0.17_{\pm0.03}$ | $0.369_{\pm0.093}$ | $0.213_{\pm0.006}$ |
| TVAE | $0.834_{\pm0.013}$ | $0.934_{\pm0.007}$ | 1.000 | $0.131_{\pm0.026}$ | $1.000_{\pm0.000}$ | $0.06_{\pm0.01}$ | – | **0.31**$_{\pm0.001}$ | $0.03_{\pm0.002}$ | **0.09**$_{\pm0.003}$ | **0.907**$_{\pm0.003}$ | $0.219_{\pm0.003}$ |
| CTAB-GAN | – | – | – | – | – | – | – | – | – | – | – | – |
| CTAB-GAN+ | – | – | – | – | – | – | – | – | – | – | – | – |
| TabDDPM | **0.985**$_{\pm0.003}$ | $0.973_{\pm0.006}$ | 1.000 | **0.880**$_{\pm0.072}$ | **0.697**$_{\pm0.074}$ | $0.21_{\pm0.01}$ | – | $0.39_{\pm0.01}$ | $0.08_{\pm0.02}$ | $0.18_{\pm0.01}$ | $0.362_{\pm0.011}$ | $0.108_{\pm0.010}$ |
| TabSyn | $0.976_{\pm0.002}$ | **0.975**$_{\pm0.001}$ | 1.000 | $0.839_{\pm0.019}$ | $0.822_{\pm0.020}$ | $0.23_{\pm0.02}$ | – | $0.39_{\pm0.02}$ | $0.09_{\pm0.03}$ | $0.19_{\pm0.03}$ | $0.340_{\pm0.060}$ | $0.080_{\pm0.035}$ |
| *CATE estimators (point estimate of $\tau(x)$ only; narrower task — reference ceiling, not direct competitors)* | | | | | | | | | | | | |
| T-learner | – | – | – | – | – | – | $0.20_{\pm0.00}$ | $0.32_{\pm0.00}$ | $0.01_{\pm0.00}$ | $0.11_{\pm0.00}$ | $0.054_{\pm0.000}$ | – |
| DR-learner | – | – | – | – | – | – | $0.15_{\pm0.00}$ | $0.32_{\pm0.00}$ | $0.01_{\pm0.00}$ | $0.11_{\pm0.00}$ | $0.051_{\pm0.001}$ | – |
| CausalPFN | – | – | – | – | – | – | $0.16_{\pm0.00}$ | $0.32_{\pm0.00}$ | $0.01_{\pm0.00}$ | $0.13_{\pm0.00}$ | $0.051_{\pm0.00}$ | – |
| DiffPO | – | – | – | – | – | – | $0.20_{\pm0.01}$ | $0.37_{\pm0.01}$ | $0.03_{\pm0.03}$ | $0.12_{\pm0.01}$ | $0.050_{\pm0.006}$ | – |

