# OpenReview forum: "Implicit Reward Alignment For Training Causally-Coherent Tabular Data Generators"
_ICML.cc/2026/Workshop/FMSD — FMSD @ ICML 2026 Poster_

### Official Review · Reviewer_7voH · 2026-05-18
**Promising Causal Alignment for Queryable Tabular Generation**

**Rating:** 7
**Confidence:** 3

**Review:**

**summary**

This paper proposes CRAFT, a reward-alignment method for improving the intervention consistency of queryable tabular generators. Using PPO with both realism and causal rewards, CRAFT improves counterfactual-oriented metrics such as counterfactual MAE, PEHE, and sign agreement over several realism-driven and queryable baselines, with especially clear gains in the linear-noiseless setting. An ablation against realism-only PPO further supports the usefulness of the causal reward.

**strength**

1. The paper tackles an important problem: tabular generators should ideally be not only realistic, but also queryable and intervention-consistent for counterfactual use cases.
2. The proposed method is conceptually simple and appealing. Using implicit reward signals to encourage causal consistency without explicitly specifying a causal graph is a clean and interesting idea.
3. The experimental study is reasonably thorough, including multiple settings from controlled semi-synthetic SCMs to a more realistic benchmark, as well as oracle/proxy reward variants and meaningful ablations.
4. The results provide useful evidence that adding a causal reward improves intervention-consistent outcome generation over strong queryable baselines such as GReaT, while largely preserving realism.

**Areas for Improvement**

1. For the counterfactual interpretation in the Twin setting to be compelling, it would be helpful to show that the binary treatment assignment contributes nontrivially beyond shared covariates X. If outcomes can largely be predicted from X alone, then the benchmark may be testing pair-level risk prediction more than genuine treatment-dependent counterfactual reasoning.
2. In the Twins setting, some realism-oriented baselines achieve lower PEHE than CRAFT, but the paper does not discuss this in detail. Since PEHE is one of the key causal metrics, it would be helpful to clarify why a realism-focused model can perform better here, and what this implies about the interpretation of counterfactual queryability in this benchmark.
3. Some notation and evidence references are not explicit enough. For instance, CRAFT-O / CRAFT-P are not clearly introduced in the main text, and some claims in Sec. 3.2 are not directly linked to the tables/figures that support them (e.g., the oracle-vs.-proxy discussion, the treatment-effect recovery discussion, and the Nonlinear-Noisy setting). More explicit pointers would improve readability

**Detailed Comments**
See points 1-3 in Areas for Improvement.

**Justication of Score**

I find the paper interesting and relevant for the workshop. Its core contribution is to show that causal reward alignment improves intervention-consistent conditional outcome generation for queryable tabular models, while largely preserving realism. The empirical study is reasonably thorough and includes oracle/proxy and ablation comparisons. My main reservations are about the interpretation of the Twins setting and a few presentation issues, but overall I believe the paper is a solid workshop contribution.

---

### Official Review · Reviewer_mR4m · 2026-05-22
**An innovative RL-approach to tabular data generation**

**Rating:** 7
**Confidence:** 3

**Review:**

**Summary**

This paper introduces Causal Reward Aligned Fine-Tuning (CRAFT), a reinforcement learning framework that fine-tunes large language models (LLMs) to generate causally coherent tabular data. By utilizing a dual-reward system comprising a realism critic and a causal evaluator (using either oracle or proxy estimators), the authors aim to align the generator's counterfactual queryability without explicitly modelling a causal graph.

**Strengths**

* **Novel Application:** Adapting RL methodologies to enforce causal consistency in tabular data generation is a highly original and relevant approach for this workshop.


* **Deployability:** The demonstration that proxy rewards (estimated from data) can recover similar benefits to oracle rewards makes the framework theoretically highly practical for real-world use cases where the underlying causal mechanisms are unknown.


* **Maintained Fidelity:** The alignment process successfully maintains the observational distribution realism of the generated data.



**Areas for Improvement**

* **Contradictory Real-World Results:** The primary weakness is that the gains of CRAFT are completely unclear in a real-world setting, severely undermining the paper's practicality claims. In Table 2, on the real-world Twins dataset, standard "Realism-only" baselines actually outperform CRAFT-P. Specifically, TVAE and TabSyn achieve better individual treatment effect recovery ($\sqrt{\text{PEHE}}$ of 0.39 and 0.34, respectively, compared to CRAFT's 0.43) and better or equal directional fidelity (Sign Agreement).
One of the paper's main claims is that optimizing purely for distributional realism is insufficient for causal consistency, thereby justifying the need for CRAFT's complex reinforcement learning pipeline. However, Table 2 shows that on the only non-synthetic, real-world dataset realism-only models actually capture the true causal dynamics ($\sqrt{\text{PEHE}}$ and Sign Agreement) better than CRAFT, which undermines the author's claim.

**Detailed Comments**

* **Different Backbone:** Since the authors claim the method is backbone-agnostic, it would be interesting to see the same recipe applied on a different LLM

**Justification of Score**
Even though the gains are not universal across all evaluations, the paper still proposes an innovative approach that shows promising results, and it would be a valuable piece of work to share more widely with the community.